# Effect of Gravity on the Scale of Compliant Shells

**DOI:** 10.3390/biomimetics5010004

**Published:** 2020-01-27

**Authors:** Victor Charpentier, Sigrid Adriaenssens

**Affiliations:** Department of Civil and Environmental Engineering, Princeton University, Princeton, NJ 08544, USA; sadriaen@princeton.edu

**Keywords:** large displacements, shell structures, morphing, gravity, compliance, scale

## Abstract

Thin shells are found across scales ranging from biological blood cells to engineered large-span roof structures. The engineering design of thin shells used as mechanisms has occasionally been inspired by biomimetic concept generators. The research goal of this paper is to establish the physical limits of scalability of shells. Sixty-four instances of shells across length scales have been organized into five categories: engineering stiff and compliant, plant compliant, avian egg stiff, and micro-scale compliant shells. Based on their thickness and characteristic dimensions, the mechanical behavior of these 64 shells can be characterized as 3D solids, thick or thin shells, or membranes. Two non-dimensional indicators, the Föppl–von Kármán number and a novel indicator, namely the gravity impact number, are adopted to establish the scalability limits of these five categories. The results show that these shells exhibit similar mechanical behavior across scales. As a result, micro-scale shell geometries found in biology, can be upscaled to engineered shell geometries. However, as the characteristic shell dimension increases, gravity (and its associated loading) becomes a hindrance to the adoption of thin shells as compliant mechanisms at the larger scales-the physical limit of compliance in the scaling of thin shells is found to be around 0.1 m.

## 1. Introduction

Thin shells, whether stiff or flexible, are curved solids with two large dimensions and a third one that is very small (thickness). In contrast to plates whose initial configuration is planar, shells are controlled by geometry and defined by their curvature. Shells built for stiffness are designed to maximize the material efficiency and reduce the overall weight-to-span ratio. By choosing an appropriate geometry for the given boundary conditions and given load case, a stiff shell experiences mostly membrane forces that can be resisted by using little material. The displacements resulting from the applied loads are practically nonexistent. In contrast, shells built for flexibility use geometry and inextensibility of materials to convert bending stresses into tuned, reversible large displacements. The use of shells as mechanisms is part of the broader, growing trend in compliant mechanisms to deform a large portion of a structure to produce movements (distributed compliance) [1,2,3,4,5,6] instead of lumped compliant hinges or common rigid body hinges.

However, when it comes to bio-inspiration, the question of scalability of natural structures becomes common. For instance, a closed shell, such as an avian egg, can rest on a plane without being damaged at a small scale. When scaled up, the shell’s self-weight, and thus the impact of gravity, increases. Under the same support conditions as the small-scale structure, the large upscaled shell could be subject to localized deformation such as buckling [7]. While this action of gravity is easily understood in this example of an egg-like shell, the question remains open to determine at which scale the action of gravity becomes too great for compliant shells to operate reliably. Shell structures span over 10 orders of magnitude across both biology and engineering. Shell mechanics are used to describe the large shape transitions of viruses [8], and of red blood cells [9], and are the mechanical system for some of the fastest repeatable plant movements [1]. Their flexibility allows the movement in engineered compliant structures such as hingeless joints [10], or active piezoelectric actuators [11]. All these flexible structures are elastically deformed, which makes them susceptible to undergo large stresses. However, similarly to their stiff counterparts, the geometry of compliant shells influences the magnitude of those stresses [12]. With advances in the modeling of large deformations [13,14], the use of shells as mechanisms is now made possible. While biologically compliant shells appear at the smallest of the 10 length scales cited above, the use of those structures in biology has started to inform the design of engineered mechanisms at larger scales. Examples of flexible shells observed in nature have inspired engineered scale adaptive structures [15,16,17,18] at the meter scale, but the question of whether such structures could be scaled up even further still remains open and drives this study. This succinct literature review shows that there is a clear gap of knowledge as to what the limiting scale of compliant shells is and what the defining parameter is that determines this scale.

The main hypothesis guiding this study is that the lack of large-scale compliant shells is due to the limiting effect of gravity-induced body forces on the shell’s movements. Therefore, the goal of this paper is to gain insight in the influence of gravity-induced forces on the ability of shells to perform as mechanisms through an order-of-magnitude approach. To achieve this goal, we are guided by the following four research tasks. First, we identify and catalogue the dimensions and mechanical characteristics of shell instances across 10 orders of magnitude of span. Second, we apply the non-dimensional Föppl–von Kármán number [19] to each of those shells to characterize the most likely deformation mode (i.e., bending or stretching). Third, in order to characterize the influence of gravity forces on a shell, we introduce a new non-dimensional number called the gravity impact number (*G_i_*), which is the ratio of the elastogravity length scale [20,21] to the characteristic dimension of the shell. The elastogravity length scale determines the limit at which bending deformations due to gravity appear in the shell. Finally, using this newly introduced parameter, we measure the scale at which compliant shells become highly susceptible to gravity induced deformations.

This paper presents an order-of-magnitude study on the behavior of shells. The main parameter used to describe the scale of the shells is their characteristic dimension (*R*), also referred to as span in engineering practice. The thickness (*H*) of the shells is related to *R* by the definition of thin and thick shells. The definition of a shell comes from the ratio of span to thickness. This ratio is found between 8 and 20 for thick shells and between 20 and 100,000 for thin shells [22]. Other geometric parameters can be used to measure the order-of-magnitude of the size. The radius of curvature or the length of a geodesic line could also be used in the context of this study. However, two reasons led to the choice of the span as controlling scale parameter: both the radii of curvature and the length of geodesic line are, in most cases, of the same order of magnitude as the span of shell, and the characteristic dimension is much more accessible in scientific literature than the two other parameters.

The main contributions of the paper are (1) the cataloguing of stiff and compliant shells across scales of 10 orders of magnitude, (2) the use the Föppl–von Kármán number to characterize the mechanical behavior of those shells, (3) the introduction of the gravity impact number to describe the scale at which the pull of gravity becomes a dominant factor in the analysis of shells, and (4) the determination that this scale is ~0.1 m.

In Section 2, the typologies of shells are introduced. In addition, the assumptions and calculation methodologies for each non-dimensional number are presented. In Section 3 of the paper, the respective values of the Föppl–von Kármán number and the gravity impact number are reported for each category of shells. In Section 4, the results are discussed, and further analysis is provided to understand the trends in data from Section 3. In Section 5, conclusions are drawn from the paper and recommendations for the use of shells as mechanisms are provided. The complete list of shells used in the study is documented in the appendices. In addition to providing the list of all the structures included, the appendices also list the principal dimensions as well as the material properties of each one of the shell instances.

## 2. Methodology

In this section, the typologies of shells featured in the study are presented and the assumptions for the non-dimensional numbers used to characterize their mechanical behavior are detailed. The comparison of the shells is done using two non-dimensional numbers: the Föppl–von Kármán number and the gravity impact number. In this section, the hypotheses for the use of each number is presented. The Föppl–von Kármán number, *γ_FvK_*, describes the type of deformation that dominates the mechanical behavior of a thin shell. The gravity impact number (*G_i_*) characterizes the influence of the gravitational force on the shell. The thin shells are considered isotropic elastic in this study.

### 2.1. Sample of Fixed and Compliant Shells

Across 10 orders of magnitude, 64 shell instances are selected from an extensive literature review that reported their mechanical behavior (Figure 1). The dimensions and references to all 64 shells are recorded in the section supplemental material (Appendix A). This study excluded several large scale compliant shells [23] for space applications since the forces due to gravity is reduced around earth’s orbit. The five categories of shells are:
Engineered stiff shells [15]. 25 large scale reinforced concrete (high Young’s modulus) thin shells used in buildings and architecture, their shape is fixed and can carry external applied loads. Their characteristic dimension (R) is in the $\left[6\times 10^{{}^{\circ}}\;m;\;8\times 10^{1}\;m\right]$ range, while their thickness H is in the $\left[5\times 10^{-2}\;m;4\times 10^{-1}\;m\right]$ rangeEngineered compliant shells [12,16,17,18,24,25,26,27,28,29,30,31,32]. 18 shells designed for use as mechanisms, they are very flexible. Materials are varied but all have high Young’s modulii. R in $\left[2\times 10^{-2}\;m;8\times 10^{0}\;m\right]$ and $H\;\mathrm{in}\;\left[1.2\times 10^{-4}\;m;9\times 10^{-4}\;m\right]$Plant compliant shells [33,34,35]. 8 plant structures that can be described as thin shells and exhibit fast and repeated motions. The material is a living tissue of low Young’s modulus ($\sim 10^{6}\;N/m^{2}$). R in $\left[1.5\times 10^{-4}\;m;1\times 10^{-2}\;m\right]$ and $H\;\mathrm{in}\;\left[3\times 10^{-5}\;m;4\times 10^{-4}\;m\right]$Avian egg stiff shells [36,37,38]. 8 stiff bird eggshells. The geometry is rigid and the material is carbon silicate of various mechanical properties detailed in [38]. R in $\left[3\times 10^{-2}\;m;1.55\times 10^{-1}\;m\right]$ and $H\;\mathrm{in}\;\left[2.2\times 10^{-4}\;m;2.55\times 10^{-3}\;m\right]$Micro-scale compliant shells [8,19,39,40,41,42]. 5 types of shells from red blood cells to virus. They have been described mechanically as a shell and deform significantly in operation. They are highly flexible. R in $\left[2\times 10^{-8}\;m;5\times 10^{-4}\;m\right]$ and $H\;\mathrm{in}\;\left[2\times 10^{-9}\;m;1\times 10^{-6}\;m\right]$

### 2.2. Quantification of Bending versus Stretching Deformation

Shells used as mechanisms rely on the property of very thin curved bodies to deform without distortion of their surface metric, i.e., without stretching. This type of deformation known as inextensional bending, or isometric bending [7,43,44] minimizes the strain energy because it does not stretch the material. The minimization of elastic strain energy is the core energy objective of compliant mechanisms. Smooth deformations without stretching are geometrically possible if the shell has free edges (and only exceptionally if the surface is closed) [44]. For thin shells, the strain energy density W includes stretching and bending. Bending in this context includes both bending and torsion. The stretching energy density $W_{stretching}$ is proportional to the shell’s thickness *H*, while the bending energy density $W_{bending}$ is proportional to the cube of the shell’s thickness *H*^3^ [45]. For equal energy levels, bending deformations can be much larger than stretching deformations. Therefore, bending allows the shell to deform with less impact on the overall elastic strain energy compared to stretching. Since isotropic thin shells are considered, the general form for the surface strain energy density W is given by the equation [45]
(1)$$W=W_{streching}+W_{bending}$$
with the stretching and bending energy densities defined as
(2)$$W_{stretching}\sim \frac{YH}{\left(1-\nu^{2}\right)}\;\epsilon^{2}$$
(3)$$W_{bending}\;\sim \;\frac{YH^{3}}{12\left(1-\nu^{2}\right)}\;\kappa^{2}\;$$
where  is Young’s modulus, ν is Poisson’s ratio, ϵ is average in-plane strain, and κ is average variation of shell curvature.

In order to measure and compare the tendency of bending-only-deformations in thin shells, the dimensionless *γ_FvK_*, number [19] is used. This number measures the ratio of stretching to bending strain energy densities and is given by
(4)$$\gamma_{FvK}=\frac{YHR^{2}}{D}$$
with R the characteristic length of the thin shell (in general of same order of magnitude as the principal curvature radii [46]) and D the bending modulus (also called the flexural stiffness),
$$D=\frac{YH^{3}}{12\left(1-\nu^{2}\right)}$$
such that after simplifications the $\gamma_{FvK}$ number becomes proportional to $R^{2}/H^{2}$
(5)$$\gamma_{FvK}=\frac{YHR^{2}}{D}=12\left(1-\nu^{2}\right)\frac{R^{2}}{H^{2}}$$

The *γ_FvK_* number predicts the type of deformation a shell will experience. Very large values of *γ_FvK_* indicate that the shell behaves similarly to a membrane. Such shells accommodate elastic compressive strain by wrinkling and if very thin, crumpling [47]. The shells with high values of *γ_FvK_* exhibit large bending and low stretching. Lower values of *γ_FvK_* correspond to thicker shells that have a high bending stiffness. Such shells have both bending and stretching deformations and require large applied loading to be deformed. The ideal behavior for a thin shell used as a mechanism is characterized by a small actuation force that results in both bending deformations and preservation of the smoothness of the surface (i.e., no crumpling or wrinkling). This force can only be made small if it activates is low stiffness deformation mode of the shell [16]. This ideal behavior occurs if the shell is stiff enough to have a bending stiffness that preserve the continuity of the material under loading and flexible enough to allow large elastic out-of-plane deformation. The instances of compliant shells selected in this study fit this description, their *γ_FvK_* values can be considered as characteristic for compliant shells. The range of *γ_FvK_* values in this study is 10^3^ to 10^8^ (see Section 3). However, physically, the maximum value of the *γ_FvK_* number is *γ_FvK_*
$\approx 10^{14}$. This value does not occur for shells since it describes the behavior of a 200 μm square sheet of graphene [48]. Graphene is one-atom thick membrane with high in plane Young’s modulus (Y=500 GPa). It has no bending stiffness and therefore is not relevant for this study. Since there is nothing thinner than a single layer of atoms, graphene constitutes the limit of physically feasible structures. 

### 2.3. Influence of Gravity Body Forces on Shells

To establish the scale limits for compliant shell, the force resulting from gravity needs to be considered as a limiting factor for their movement. A non-dimensional number, called the gravity impact, *G_i_*, number is introduced in this paper to quantify the gravitational impact on a shell’s behavior. The *G_i_* number is defined as the ratio of the elastogravity length scale $l_{eg}$ [20,21] to the characteristic dimension of the shell R. 

The gravitational potential energy density ($W_{gravity}$) scales as
(6)$$W_{gravity}\;\sim \;g\rho \delta^{2}$$

With *g* being the acceleration of gravity, ρ being the volumetric mass density of the material, and δ being the deformation due to gravity. From a dimensional point of view, the variation of average curvature κ can be expressed as a function of δ as $\kappa \;\sim \;\delta /R^{2}$. The gravitational pull causes the shell to bend when the bending energy and the gravitational potential energy are of the same order of magnitude, i.e., $W_{bending}\approx W_{gravity}$. This situation occurs for $R\;\sim \;l_{eg}$. Equating Equations (3) and (6) yields Equation (7)
(7)$$l_{eg}\;\sim \;{\left(\frac{D}{g\rho }\right)}^{1/4}$$

Therefore, the nondimensional *G_i_* number for a thin shell is
(8)$$Gi=\frac{l_{eg}}{R}={\left(\frac{D}{g\rho R^{4}}\right)}^{1/4}$$

Therefore, if *G_i_* is larger than one, the characteristic dimension of a thin shell is smaller than $l_{eg}$: the gravity effect on the behavior of the compliant shell can be ignored. The nondimensional *G_i_* number determines the tendency of a compliant shell to be affected by the gravitational pull as a function of its scale. Values of *G_i_* lower than unity indicate that gravitational forces exert a large influence on the shell’s mechanical behavior. In contrast, *G_i_* values over one indicate gravitational forces are not of key importance in the deformation. The gravitational pull increases as the characteristic dimension of the shell increases. Compliant thin shells of large dimensions are rare but there are some examples of such shells where the characteristic dimension is in the order of magnitude of $10^{0}$ m or below. Therefore, the *G_i_* number is used in this study to detect and highlight the scaling limits of compliant thin shell.

## 3. Results

### 3.1. Föppl–von Kármán Number Values Across Scales

The 64 thin shells included in this study are plotted by thickness *H* and characteristic dimension *R* in Figure 2. *γ_FvK_* describes whether stretching and/or bending deformations control the deformed state of the shell. Being a non-dimensional number, it applies to any shell, independent of the magnitude of its characteristic dimension *R*. The average values of *γ_FvK_* shown in Table 1 are within the range $10^{3}$ to $10^{8}$. To understand the variability observed in Table 1, we need to define precisely the subcategories of solids that appear on Figure 2. In this study the ratio *R*/*H* for a thin shell is adopted from [22] and given by
(9)$$20\leq \;\frac{R}{H}\leq 100,000$$

In comparison, thick shells have a larger *R/H* ratio also defined in [22] and given by
(10)$$8\leq \;{\frac{R}{H}}_{thickShell}\;\leq 20$$

All stiff shells and compliant engineered shells in this study fall within the range of *R/H* ratios defined in Equations (9) and (10), respectively. The 3D solids with *R/H* values less than 8 are rigid bodies that cannot be described as having two spatial dimensions much larger than the third one. Therefore, they are not considered shells and Equations (3) and (9) do not apply to them, those structures appear on the left-hand side of Figure 2. In contrast, when the *R*/*H* ratio is larger than 100,000, shells become extremely thin. They lose any bending stiffness and can only experience in-plane forces (stretching). They can no longer be called shells and are referred to as membranes. (right-hand side of Figure 2). The average values of *γ_FvK_* in Table 1 indicate a mechanical behavior dominated by bending deformation for both stiff and compliant shells. Overall, since thin shells have *R/H* ratios in the range of [20;100,000], their *γ_FvK_* values are bounded by lower $(\gamma_{FvK}\;\sim 10^{3}$) and upper bounds $(\gamma_{FvK}\;\sim 10^{8}$). This observation indicates that thin shells—whether they are engineered stiff or compliant, plant compliant, micro scale compliant, or egg stiff—exhibit similar mechanical behavior, which is dominated by bending deformations across scales.

Of the 64 thin shell typologies recorded in this study, 95% have values of *γ_FvK_* between $10^{3}$ and $10^{8}$, as shown in Figure 2 and Figure 3. In the sample of shells selected for this study, only some of the compliant plant shells present values of *γ_FvK_* lower than 10^3^ (Figure 3). Those same instances are on the border of the range of *R*/*H* ratios that characterizes shell structures (Figure 2). The main simplifying hypothesis of this study is that the material of the structures selected is isotropic elastic. In the case of the plant structures, the complex nature of the plant material (referred to as plant tissue) requires further justification for being included in this study. Biological tissues that constitute the moving organs of the plants instances included in the study are a hierarchized, non-homogeneous material [1]. As a living material, not all parts of tissue perform structural functions [49]. The structural layers of the tissue are thinner than the overall tissue [1] therefore in the cases presented in the study, the ratio *R*/*H* of the plants despite being loaer than other examples of shells are still accepted. 

A specific example of this behavioral similarity can be found in the Algeciras Market Hall’s reinforced concrete shell [15] and red blood cells. Both of those structures have similar Föppl–von Kármán number $\gamma_{FvK}\sim 10^{5}$, which indicates a similar tendency to bending deformation over stretching deformations for both structures. This high *γ_FvK_* indicates a high in-plane stiffness compared to the out-of-plane bending stiffness. Therefore, bending deformations are more likely to occur than stretching for both structures. In theory, the structure of the Algerciras Market hall should be able to undergo similar reversible large shape changes as red blood cells. The market hall is a stiff concrete shell considered a model of shell design [15]. While in pure mechanical terms the concrete structure could be used as a compliant shell, the actual Algerciras Market Hall is dominated by dead-load’s vertical action and subjected to edge boundary conditions. 

### 3.2. Impact of Gravity on Shell Mechanical Behavior Across Scales

The thickness and the characteristic dimension of the shells are related by the ratio *R/H* discussed in Section 3.1. Therefore, the characteristic dimension will be taken as the reference indicator of a shell’s geometry going forward. The relationship between the *G_i_* number and geometry is shown in Figure 4. The Figure shows that stiff engineered thin shells have the largest values of *G_i_*, while micro-scale compliant shells have the lowest values.

In accordance with Equation (8), the gravitational force is larger (in magnitude) as the scale of the shell increases. This is appears in Figure 4 with thin shell of larger characteristic dimensions having low values of *G_i_* such as for example façade shading shells [16,17,28]. Shells with a characteristic dimension *R* lower than 0.1 m tend to have Gi>1. For these shells, large deformation caused by gravity does not occur. The relationship Gi>1 only occurs for one-third of compliant engineered shells, which means that most engineered shells must deal with the influence of gravity. All studied stiff engineered shells have an elastogravity length scale shorter than their characteristic dimension *R*. This observation indicates that for these shells the gravitational forces due to self-weight dominate the elastic bending resistance. The average value of *G_i_* is found to be 0.109 for stiff engineered shells, 0.610 for compliant engineered shells, 2.465 for plant compliant shells, 0.822 for the egg shells, and 7.739 for the micro-scale compliant shells.

In addition, shells with a characteristic dimension larger than 0.1 m consistently have values of *G_i_* lower than one (Figure 4). This scale is displayed by the red dotted line on Figure 4. No structure to the right of this line has a gravity impact number larger than one.

There is not a clear division defined by *G_i_* between compliant and stiff thin shells. Some engineered compliant thin shells are used as mechanisms but have a lower *G_i_* value than the one of stiff shells. A high value of *G_i_* can also indicate a shell with large thickness *H* with a corresponding low *γ_FvK_* value. The plant compliant shells have relatively high *G_i_* values, which means the shell does not deform under the influence of gravity. The larger plant compliant shells have *G_i_* values comparable to those of stiff engineered shells, indicating that the shell would be susceptible to the influence of gravity. However, for the living tissues, the ratio of volumetric mass density *ρ* to Young’s modulus *Y* is $\sim 10^{3}$ times lower than that for engineered shells, which explains some of the low values of *G_i_* despite the small characteristic dimensions *R*.

## 4. Discussion

Thin shells have a tendency to deform in bending rather than in stretching across scales. Most thin shells studied have values of the Föppl–von Kármán number *γ_FvK_* between $10^{3}$ and $10^{8}$. This non-dimensional number is significant because it unifies the behavior of shells across scales. This outcome is in line with the bio-inspiration approach that distills geometries of a plant or micro-scale shell and scales them up for engineering applications [1,16]. As long as the ratio of characteristic dimension *R* over thickness *H* is kept high, the mechanical behavior of the compliant shell is similar at the large engineered scale and the observed biological scale. The five categories of thin shells presented in this paper (i.e., engineered stiff, engineered compliant, plant compliant, micro-scale compliant, and egg stiff) have instances with $\gamma_{FvK}$ in the range of $10^{4}$ to $10^{5}$. This observation exemplifies the that shells used as mechanisms appear at all scales Figure 5. Thin shell can have a similar mechanical behavior dominated by bending deformation across 10 orders of magnitude of their characteristic dimension *R*.

Compliant shells found in plants are made from living tissues, a multileveled arrangement of cells. In this material, the transport of water generated by electro-chemical reactions increases the water pressure in select part of the tissue, thus generating actuation [1]. Plant tissues are a self-actuating material. As discussed in Section 3.1, despite being classified as thick shell or almost 3D solids, plant mechanisms still feature in this study due to the properties of the thin structural layers of the motile plant organs being the main structural component of the mechanisms. For plants, the elastogravity length *l_eg_* scale is large compared to their characteristic dimension *R*. In the genus *Stylidium* for example, the characteristic dimension *R* of the mechanism is 4.3 times larger than the elastogravity length scale *l_eg_*, which indicates that the plant’s movement is quasi unaffected by gravity. In general, plants can move without having the deformed geometry being influenced too much by gravity. The orientation of their mobile parts with respect to the gravitational pull does not obstruct or favor the shell movement.

Large-span stiff engineered shells use engineered materials with high Young’s moduli *Y* and are designed to have a fixed shape that minimizes bending stresses and can thus be made very thin. In contrast, compliant shells must repeat elastic deformations reliably at a low actuating cost. The scale at which shells’ compliant deformations start to be affected by gravity is R ~ 0.1 m. Below that scale, compliant shells operate independent of gravity. For instance, the adaptive air inlet for aeronautic applications described in [12] must be able to function under any orientation of the airplane. In contrast, a compliant shell for adaptive shading of buildings does not have the same constraints as it operates in a position aligned with gravity’s vertical orientation and can therefore be up scaled to larger sizes [16,17,18].

## 5. Conclusions

When upscaling stiff and compliant shells from small scale biology to large scale engineered applications, the pull of gravity needs to be accounted for. The first contribution of this paper is the identification and logging of the dimensions and mechanical characteristics of 64 shell instances across 10 orders of magnitude of span. The shells listed are drawn from micro-biology, plant biology, animal biology, and engineering. Using the non-dimensional Föppl–von Kármán number, bending was shown to be more likely to occur than stretching as the dominating deformation mode for shells across all scales. Stiff engineered shells are shaped so that this tendency is neutralized but compliant shells take advantage of it to deform (second contribution). In order to characterize the influence of gravity on those compliant shells across scales, *G_i_* was introduced in this paper. This non dimensional number determines at what scale gravity becomes relevant in the study of shell mechanics (third contribution). In particular, *G_i_* is defined as the ratio of the elastogravity length scale to the characteristic dimension of the shell and measures whether the scale at which bending deformation due to self-weight appears in a shell is larger or smaller than the actual size of the shell. The fourth contribution is the identification of the scale at which shells become influenced by gravity. Based on the characteristics of the 64 listed shells and using *G_i_*, it is shown that the effect of gravity on compliant shells sets on at a scale of ~0.1 m. Compliant shells at larger scales (*R* > 0.1 m) are prone to self-weight deformation under gravity load. This deformation can hinder their function depending on the nature of the application. A mechanism based on compliant shells that needs to perform reliably under varying orientation (e.g., airplane wing) will not be able to be scaled to large scales. However, if the application does not demand a change of orientation, the structure can be scaled up providing that the orientation of gravity is taken into account in the design of the compliant shell.

The following strategies can be used for compliant thin shells to circumvent Gi<1 while having a high *γ_FvK_*.
The mobile part of the shell is designed to possess enough stiffness to be cantilevered. In most compliant shell building shading systems [16,17,18], this stiffness is provided by curvature and built-up stresses.The mechanism is oriented to limit the increase of cantilevered length during the movement. For example, the façade of the Yoesu Expo 2012 Pavilion was designed so that the flexible shell elements do not create large overhangs during the out-of-plane buckling deformation [28]. The longest elements are 8 m tall and still able to be elastically deformed repeatedly.The bending deformation of thin shells can be predicted by studying the possible isometric deformation of their geometry [44]. The deformation of very curved surfaces could lead to mechanisms being able to withstand gravity better due to their doubly-curved geometry [26].The final strategy to create large-scale compliant thin shells is to operate in outer space. The behavior of shells is similar across scales. Bending deformation modes dominate stretching modes when shells are thin enough. Being able to remove gravity forces could lead to large shells being used as compliant mechanisms.

Determining the limits of scalability of shell used as mechanisms will allow designers to expand the use of those structures, using the geometry of small-scale thin shells for inspiration in engineered applications. To support this development, further work should be carried on form finding applied to large displacements of shells. Throughout this paper, parallels have been made between the design of stiff shells and the design of compliant shells. There is a well-established field of structural research dedicated to the search for optimal forms for stiff shells. The development of tools for finding appropriate and custom flexible shell geometries could lead to structural designers departing from a pure mimetic based approach that upscales geometries to a generative approach that is able to generate appropriate flexible shapes for kinematic problems.

## Figures and Tables

**Figure 1 biomimetics-05-00004-f001:**
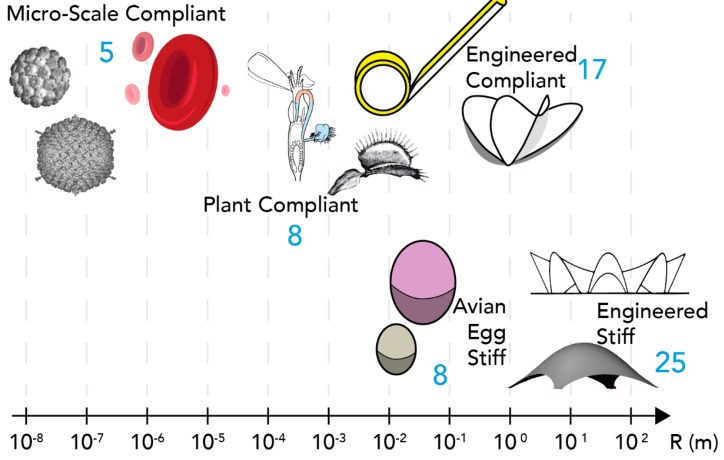
Varying size/scale (R) of the five types of shells included in the study. The number of typologies for each type of shell is shown in blue.

**Figure 2 biomimetics-05-00004-f002:**
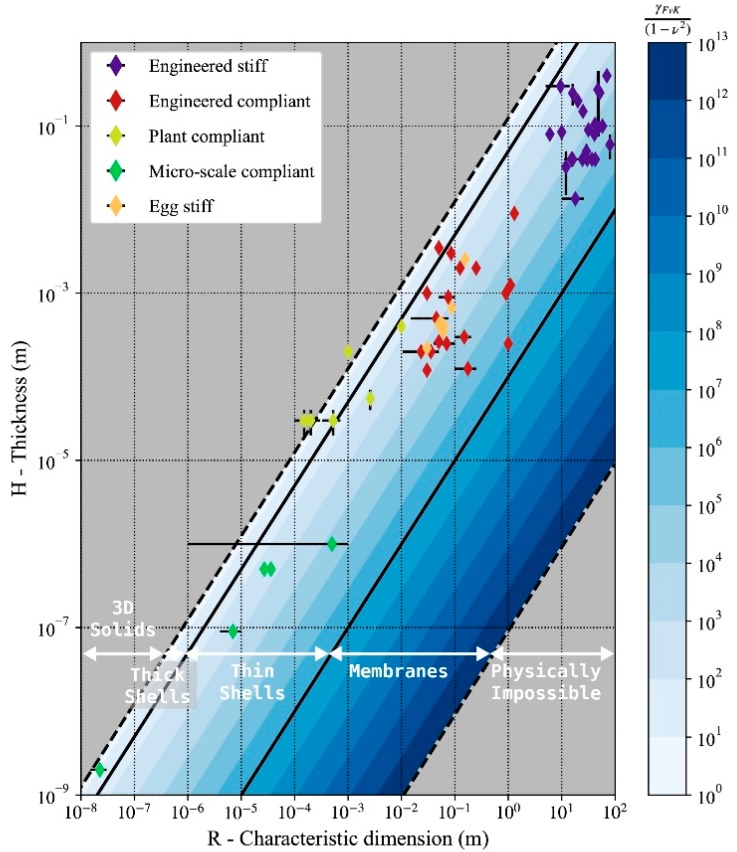
Geometric properties and *γ_FvK_* values for stiff and compliant thin shells, plant compliant thin shells, and compliant micro-scale shells. The scale for both axes is logarithmic.

**Figure 3 biomimetics-05-00004-f003:**
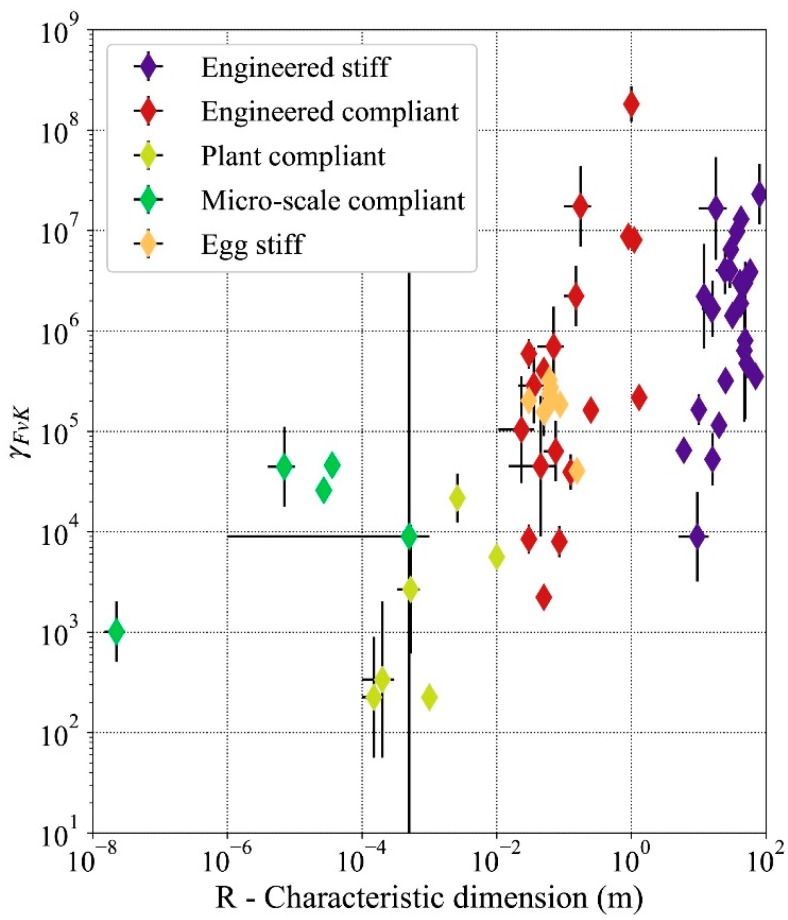
Föppl–von Kármán number, $\gamma_{FvK}$, in thin shells as a function of the characteristic dimension. The scale for both axes is logarithmic.

**Figure 4 biomimetics-05-00004-f004:**
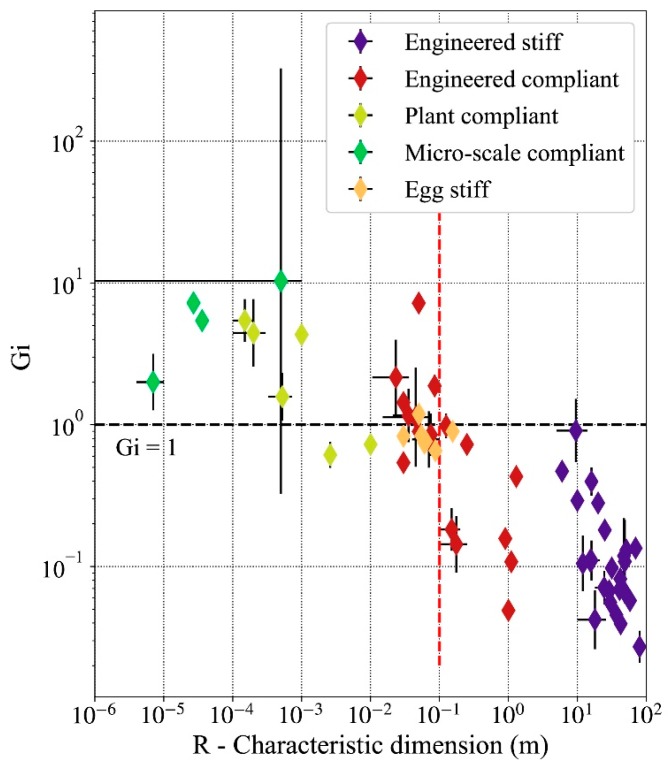
Gravitational force density impact in thin shells as a function of the characteristic dimension. The scale for both axes is logarithmic. The horizontal dotted line indicates values *G_i_* = 1 for which the gravitational force becomes predominant in the equilibrium of the shell. The red dotted line at *R* = 0.1 m represent the approximate limit at which thin shells start to be constrained by gravity.

**Figure 5 biomimetics-05-00004-f005:**
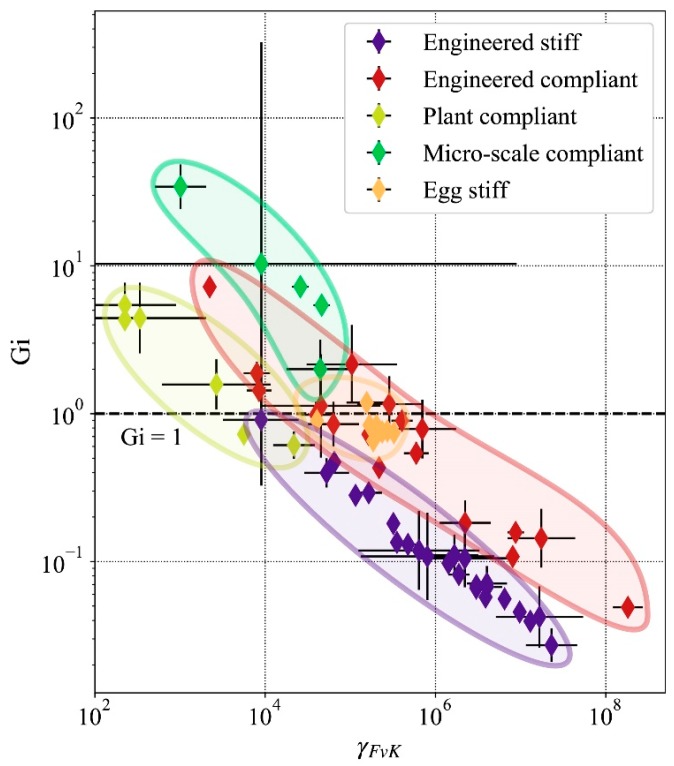
Classification of compliant and stiff thin shells. The nondimensional gravitational force density *G_i_* is plotted as a function of the Föppl–von Kármán number *γ_FvK_*. The dotted line indicates values *G_i_* = 1 for which the gravitational force becomes predominant in the equilibrium of the shell.

**Table 1 biomimetics-05-00004-t001:** Average values of *γ_FvK_* for the five types of shells.

Shell Type	$\mathrm{Average}\;\gamma_{FvK}$
Stiff Engineered	$3.95\;\times 10^{6}$
Compliant Engineered	$1.33\times 10^{7}$
Stiff Avian Egg	$1.98\times 10^{5}$
Compliant Plant	$3.84\times 10^{3}$
Compliant Micro-Scale	$2.54\times 10^{4}$

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
