# Peer review of "Effect of Gravity on the Scale of Compliant Shells"

_biomimetics, 2020, doi:10.3390/biomimetics5010004_

Round 1

Reviewer 1 Report

This work “Physical Limits of Large Displacements in the Scaling of Thin Shells” aimed to present the physical limits for compliant shell design using an analytical and data-driven approach. 64 instances of shells across length scales have been arranged into five categories being engineering stiff and compliant, plant compliant, avian egged stiff and micro-scale compliant shells. Although the work is interesting, some points should be improved:

The title of the paper is too general. It should be more specific so that readers can access it easily. The abstract of the article can be more concise and must be restructured. The main studies carried out in this research are not clearly defined in the abstract. There are numerous grammatical errors throughout the paper, which need to be correct. The literature review is ambiguous; include some more recent state of the art papers in Literature review for better understanding. Delete some extra irrelevant papers that are in the manuscript. Refer the following papers for better understanding: Salahifar, Raydin, and Magdi Mohareb. "Generalized theory for the dynamic analysis of thin shells with application to circular cylindrical geometries." Thin-Walled Structures 139 (2019): 347-361. Ferraro, Serena, and Sergio Pellegrino. "Topology Optimization of Composite Self-Deployable Thin Shells with Cutouts." AIAA Scitech 2019 Forum. 2019. Guo, Y., H. Do, and M. Ruess. "Isogeometric stability analysis of thin shells: From simple geometries to engineering models." International Journal for Numerical Methods in Engineering 118.8 (2019): 433-458. The Introduction part of the article must be revised to make it better structured for the readers. Try to explain the previous work related to different aspects of the current research and connect it with the problem statement, in the end, i.e. identifying the gap and why was this model necessary to develop. Try to write two more keywords for a better understanding of readers. The paper seems to be more like a report rather than a research paper. Try to restructure the paper. Results and discussion are poorly defined. Improve it. Prove the novelty of the paper with referred papers in the last paragraph of the introduction as it seems to be very preliminary research that is already available to the scientific community. In the last paragraph of the introduction section, mention the novelty of this paper with the previous state of the art research. The results of the model have not been elaborated in detail. In conclusion, write the result of each discussion in pointwise. Resize Fig. 3.

Author Response

Answer to reviewer 1

biomimetics-646505

In this document, the authors address the comments of reviewer 1. The authors would like to thank reviewer 1 for his/her comments aimed at improving the quality of the manuscript.

Comments from reviewer 1

The title of the paper is too general. It should be more specific so that readers can access it easily.

The authors have changed the name of the paper to make it more precise. The reviewed title is: “The effect of gravity on the scale of compliant shells”

There are numerous grammatical errors throughout the paper, which need to be correct.

The authors have taken the manuscript through a grammar check and the revised manuscript has been reviewed by a native speaker.

The literature review is ambiguous; include some more recent state of the art papers in Literature review for better understanding. Delete some extra irrelevant papers that are in the manuscript. Refer the following papers for better understanding: Salahifar, Raydin, and Magdi Mohareb. "Generalized theory for the dynamic analysis of thin shells with application to circular cylindrical geometries." Thin-Walled Structures 139 (2019): 347-361. Ferraro, Serena, and Sergio Pellegrino. "Topology Optimization of Composite Self-Deployable Thin Shells with Cutouts." AIAA Scitech 2019 Forum. 2019. Guo, Y., H. Do, and M. Ruess. "Isogeometric stability analysis of thin shells: From simple geometries to engineering models." International Journal for Numerical Methods in Engineering 118.8 (2019): 433-458.

The authors rewrote the introduction to more effectively deliver context for the study. The references suggested by the reviewer have been included and a few additional references have been included to concisely present the state-of-the-art of compliant and stiff shell mechanics. The nine new references that have been added are:

Verotti, M., A. Dochshanov, and N.P. Belfiore, A comprehensive survey on microgrippers design: Mechanical structure. Journal of Mechanical Design, 2017. 139(6): p. 060801. Sun, J., et al., Morphing aircraft based on smart materials and structures: A state-of-the-art review. Journal of Intelligent material systems and structures, 2016. 27(17): p. 2289-2312. Li, S. and K. Wang, Plant-inspired adaptive structures and materials for morphing and actuation: a review. Bioinspiration & biomimetics, 2016. 12(1): p. 011001. Zhang, Z., et al., Bistable morphing composite structures: a review. Thin-walled structures, 2019. 142: p. 74-97. Chillara, V. and M.J. Dapino, Review of Morphing Laminated Composites. Applied Mechanics Reviews, 2020. 72(1). Ferraro, S. and S. Pellegrino. Topology Optimization of Composite Self-Deployable Thin Shells with Cutouts. in AIAA Scitech 2019 Forum. 2019. Tzou, H.H., Piezoelectric Shells: Sensing, Energy Harvesting, and Distributed Control—Second Edition. Vol. 247. 2018: Springer. Salahifar, R. and M. Mohareb, Generalized theory for the dynamic analysis of thin shells with application to circular cylindrical geometries. Thin-Walled Structures, 2019. 139: p. 347-361. Guo, Y., H. Do, and M. Ruess, Isogeometric stability analysis of thin shells: From simple geometries to engineering models. International Journal for Numerical Methods in Engineering, 2019. 118(8): p. 433-458.

The Introduction part of the article must be revised to make it better structured for the readers. Try to explain the previous work related to different aspects of the current research and connect it with the problem statement, in the end, i.e. identifying the gap and why was this model necessary to develop.

First, the research gap is now formulated more clearly in the introduction:

“But when it comes to bio-inspiration, the question of scalability of natural structures becomes common. For instance, a closed shell, such as an avian egg, can rest on a plane without being damaged at a small scale. However, when scaled up the shell’s self-weight and thus the impact of gravity increases. Under the same support conditions as the small-scale structure, the large upscaled shell could be subject to localized deformation such as buckling [7]. While this action of gravity is easily understood in this example of an egg-like shell, the question remains open to determine at which scale the action of gravity becomes too great for compliant shells to operate reliably. Shell structures span over 10 orders of magnitude across both biology and engineering. Shell mechanics are used to describe the large shape transitions of viruses [8], and of red blood cells [9], and are the mechanical system for some of the fastest repeatable plant movements [1]. Their flexibility allows the movement in engineered compliant structures such as hingeless joints [10] or active piezoelectric actuators [11]. All these flexible structures are elastically deformed, which makes them susceptible to undergo large stresses. However, similarly to their stiff counterparts, the geometry of compliant shells influences the magnitude of those stresses [12]. With advances in the modeling of large deformations [13, 14], the use of shells as mechanisms is now made possible. While biological compliant shells appear at the smallest of the 10 length scales cited above, the use of those structures in biology has started to inform the design of engineered mechanisms at larger scales. Examples of flexible shells observed in nature have inspired engineered scale adaptive structures [15-18] at the meter scale, but the question of whether such structures could be scaled up even further still remains open and drives this study. This succinct literature review shows that there is a clear gap of knowledge as to what the limiting scale of compliant shells is and what the defining parameter is that determines this scale.

Second, the research goal and the four research tasks of the article have been defined more clearly:

“The main hypothesis guiding this study is that the lack of large-scale compliant shells is due to the limiting effect of gravity-induced body forces on the shell’s movements. Therefor the goal of this paper is to gain insight in the influence of gravity-induced forces on the ability of shells to perform as mechanisms through an order-of-magnitude approach. To achieve this goal, we are guided by the following four research tasks. First, we identify and catalogue the dimensions and mechanical characteristics of shell instances across 10 orders of magnitude of span. Second, we apply the non-dimensional Föppl-von-Kármán number [19] to each of those shells to characterize the most likely deformation mode (i.e. bending or stretching). Third, in order to characterize the influence of gravity forces on a shell, we introduce a new non-dimensional number called the gravity impact number (Gi), which is the ratio of the elastogravity length scale [20, 21] to the characteristic dimension of the shell. The elastogravity length scale determines the limit at which bending deformations due to gravity appear in the shell. Finally, using this newly introduced parameter, we measure the scale at which compliant shells become highly susceptible to gravity induced deformations.”

Third, the research contributions are now clearly stated:

“The main contributions of the paper are 1) the cataloguing of stiff and compliant shells across scales of 10 orders of magnitude, 2) the use the Föppl-von-Kármán number to characterize the mechanical behavior of those shells, 3) the introduction of the gravity impact number to describe the scale at which the pull of gravity becomes a dominant factor in the analysis of shells and 4) the determination  that this scale is ~0.1 m.”

Try to write two more keywords for a better understanding of readers.

The keywords have been updated and are now: large displacement, shell, morphing, gravity, compliance, scale

The paper seems to be more like a report rather than a research paper. Try to restructure the paper.

In this paper we establish what the limiting scale is for compliant shells.  To establish this scale, we introduce a gravity impact number and make an order of magnitude comparison of the mechanics based on a literature survey of 64 shell instances. To ensure that the paper reads as a research paper, we structured the paper following the standards of scientific writing (i.e. introduction, methodology section, results section, discussion and conclusions). The objective and contributions of the paper have been stated more clearly in the Introduction section. As suggested by the reviewer, the authors have improved the clarity and structure of the paper overall by re-writing the introduction, writing a precise and concise header for each section, simplifying the discussion and making sure that the conclusions are more succinct and provide suggestions for further development of the research field.

Results and discussion are poorly defined. Improve it.

The authors appreciate the reviewer’s comment and have made significant changes to the discussion and the conclusion section.

The results section now reads as:

3.1. Föppl-von-Kármán number values across scales

The 64 thin shells included in this study are plotted by thickness  and characteristic dimension  in Figure 2.  describes whether stretching and/or bending deformations control the deformed state of the shell. Being a non-dimensional number, it applies to any shell, independent of the magnitude of its characteristic dimension . The average values of  shown in Table 1 are within the range  to . To understand the variability observed in Table 1, we need to define precisely the subcategories of solids that appear on Figure 2. In this study the ratio  for a thin shell is adopted from [22] and given by

(8)

In comparison, thick shells have a larger  ratio also defined in [22] and given by

(9)

All stiff shells and compliant engineered shells in this study fall within the range of  ratios defined in equation (8) and (9) respectively. The 3D solids with   values less than 8 are rigid bodies that cannot be described as having two spatial dimensions much larger than the third one. Therefore, they are not considered shells – equations (3) and (8) do not apply to them, those structures appear on the left-hand side of Figure 2. In contrast, when the  ratio is larger than 100 000, shells become extremely thin. They lose any bending stiffness and can only experience in-plane forces (stretching). They can no longer be called shells and are referred to as membranes. (right-hand side of Figure 2). The average values of  in Table 1 indicate a mechanical behavior dominated by bending deformation for both stiff and compliant shells. Overall since thin shells have  ratios in the range ], their  values are bounded by lower ) and upper bounds ). This observation indicates that thin shells – whether they are engineered stiff or compliant, plant compliant, micro scale compliant or egg stiff – exhibit similar mechanical behavior, which is dominated by bending deformations across scales.

95% of the 64 thin shell typologies recorded in this study have values of  between  and , as shown in Figures 2 and 3. In the sample of shells selected for this study, only some of the compliant plant shells present values of  lower than 103 (Figure 3). Those same instances are on the border of the range of ratios that characterizes shell structures (Figure 2). The main simplifying hypothesis of this study is that the material of the structures selected is isotropic elastic. In the case of the plant structures, the complex nature of the plant material (referred to as plant tissue) requires further justification for being included in this study. Biological tissues that constitute the moving organs of the plants instances included in the study are a hierarchized, non-homogeneous material [1]. As a living material, not all parts of tissue perform structural functions [49]. The structural layers of the tissue are thinner than the overall tissue [1] therefore in the cases presented in the study, the ratio  of the plants despite being loaer than other examples of shells are still accepted.

A specific example of this behavioral similarity can be found in the Algeciras Market Hall reinforced concrete shell [15] and red blood cells. Both of those structures have similar Föppl-von-Kármán number , which indicates a similar tendency to bending deformation over stretching deformations for both structures. This high  indicates a high in-plane stiffness compared to the out-of-plane bending stiffness. Therefore, bending deformations are more likely to occur than stretching for both structures. In theory, the structure of the Algerciras Market hall should be able to undergo similar reversible large shape changes as red blood cells. The market hall is a stiff concrete shell considered a model of shell design [15]. While in pure mechanical terms the concrete structure could be used as a compliant shell, the actual Algerciras Market Hall is dominated by dead-load’s vertical action and subjected to edge boundary conditions.

3.2. Impact of gravity on shell mechanical behavior across scales

The thickness and the characteristic dimension of the shells are related by the ratio  discussed in Section 3.1. Therefore, the characteristic dimension will be taken as the reference indicator of a shell’s geometry going forward. The relationship between the  number and geometry is shown in Figure 4. The Figure shows that stiff engineered thin shells have the largest values of , while micro-scale compliant shells have the lowest values.

In accordance with equation (7), the gravitational force is larger (in magnitude) as the scale of the shell increases. This is appears in Figure 4 with thin shell of larger characteristic dimensions having low values of  such as for example façade shading shells [16, 17, 28]. Shells with a characteristic dimension  lower than 0.1 m tend to have . For these shells, large deformation caused by gravity does not occur. The relationship  only occurs for 1/3 of compliant engineered shells, which means that most engineered shells must deal with the influence of gravity. All studied stiff engineered shells have an elastogravity length scale shorter than their characteristic dimension . This observation indicates that for these shells the gravitational forces due to self-weight dominate the elastic bending resistance. The average value of  is found to be 0.109 for stiff engineered shells, 0.610 for compliant engineered shells, 2.465 for plant compliant shells, 0.822 for the egg shells and 7.739 for the micro-scale compliant shells.

In addition, shells with a characteristic dimension larger than 0.1 m consistently have values of  lower than one (Figure 4). This scale is displayed by the red dotted line on Figure 4. No structure to the right of this line has a gravity impact number larger than one.

There is not a clear division defined by  between compliant and stiff thin shells. Some engineered compliant thin shells are used as mechanisms but have a lower  value than the one of stiff thin shells. A high value of  can also indicate a shell with large thickness  with a corresponding low  value. The plant compliant shells have relatively high  values. The shell does not deform under the influence of gravity. The larger plant compliant shells have  values comparable to those of stiff engineered shells. For the living tissues, the ratio of volumetric mass density  to Young’s modulus  is  times lower than that for engineered shells, which explains some of the low values of  despite the small characteristic dimensions .

Prove the novelty of the paper with referred papers in the last paragraph of the introduction as it seems to be very preliminary research that is already available to the scientific community.

The context for this paper is framed in the context of the increasing body of research on compliant shell mechanisms, presented in the Introduction section. In that same section we now clearly state the scientific contributions of the paper:  “The main contributions of the paper are 1) the cataloguing of stiff and compliant shells across 10 orders of magnitude, 2) the use the Föppl-von-Kármán number to characterize the mechanical behavior of those shells, 3) the introduction of the gravity impact number to describe the scale at which the pull of gravity become a dominant factor in the analysis of shells and 4) the determination  that this scale is ~0.1 m.”

In the last paragraph of the introduction section, mention the novelty of this paper with the previous state of the art research. The results of the model have not been elaborated in detail.

We have addressed the scientific contributions of the paper in the Introduction section in the previous comment. 

In conclusion, write the result of each discussion in pointwise.

The four main contributions of the paper have now been clearly listed in the conclusion section:

“When upscaling stiff and compliant shells from small scale biology to large scale engineered applications, the pull of gravity needs to be accounted for. The first contribution of this paper is the identification and logging of the dimensions and mechanical characteristics of 64 shell instances across 10 orders of magnitude of span. The shells listed are drawn from micro-biology, plant biology, animal biology and engineering. Using the non-dimensional Föppl-von-Kármán number, bending was shown to be more likely to occur than stretching as the dominating deformation mode for shells across all scales. Stiff engineered shells are shaped so that this tendency is neutralized but compliant shells take advantage of it to deform (second contribution). In order to characterize the influence of gravity on those compliant shells across scales, Gi was introduced in this paper. This non dimensional number determines at what scale gravity becomes relevant in the study of shell mechanics (third contribution). In particular Gi is defined as the ratio of the elastogravity length scale to the characteristic dimension of the shell and measures whether the scale at which bending deformation due to self-weight appears in a shell is larger or smaller than the actual size of the shell. The fourth contribution is the identification of the scale at which shells become influenced by gravity. Based on the characteristics of the 64 listed shells and using Gi, it is shown that the effect of gravity on compliant shells sets on at a scale of ~0.1 m. Compliant shells at larger scales (R > 0.1m) are prone to self-weight deformation under gravity load. This deformation can hinder their function depending on the nature of the application. A mechanism based on compliant shells that needs to perform reliably under varying orientation (e.g. airplane wing) will not be able to be scaled to large scales. However, if the application does not demand a change of orientation, the structure can be scaled up providing that the orientation of gravity be take into account in the design of the compliant shell.”

Resize Fig. 3.

This size of the figures was reduced so that the font size of Fig.3. matches the font size of the main text.

Reviewer 2 Report

This paper is well written in english

and the results appear to be correct. 

The methods used to find the results

provide innovations in the scientific field used.

The paper is accepted for pubication in the journal.

Author Response

Answer to reviewer 2

biomimetics-646505

This document takes the remarks of reviewer 2 and provides point by point answers to them.

Comments from reviewer 2

The authors would like to thank reviewer 2 for his or her feedback.

This paper is well written in English and the results appear to be correct.

The methods used to find the results provide innovations in the scientific field used.

The paper is accepted for publication in the journal.

Reviewer 3 Report

The manuscript deals with an interesting investigation concerning the structural study in the scaling of shell-based components. Different scenarios are presented at micro-scale level showing how their potentialities can be extended to larger engineering systems. The mechanical performance improvement has been assessed as function of two non-dimensional parameters. The first one is an energetic ratio (Föppl-von-Kármán) characterizing the elastic properties while another coefficient has been implemented to assess the gravity inertial effect on the deformation.

By my side, the paper faces a very interesting topic which is expected to contribute on the field of “Bio-based Structures” treated by Biomimetics journal: the technical content of the paper is worthy of publication. The configurations studied could find interesting application in morphing systems which should have a greater flexibility along the chord-wise but good stiffness along the span-wise direction to withstand the aero-loads.

Anyway, the reviewer suggests the following comments, to be considered and addressed in order this paper is considered for publication.

Add a list comprising all the symbols and abbreviations; The introduction is still short; expand with further references and clearly define the innovative aspect of the manuscript; How are the flexibility characteristics implemented in the physical model? Explain more Have been conceptualized some experiments to validate the theories developed?

Author Response

Answer to reviewer 3

biomimetics-646505

This document takes the remarks of reviewer 3 and provides point by point answers to them.

Comments from reviewer 3

Add a list comprising all the symbols and abbreviations

The authors added a list the symbols and abbreviations before the Introduction section.

The introduction is still short; expand with further references and clearly define the innovative aspect of the manuscript;

In this paper we establish what the limiting scale is for compliant shells.  To establish this scale, we introduce a gravity impact number and make an order of magnitude comparison of the mechanics based on a literature survey of 64 shell instances. To ensure that the paper reads as a research paper, we structured the paper following the standards of scientific writing (i.e. introduction, methodology section, results section, discussion and conclusions). The objective and contributions of the paper have been stated more clearly in the Introduction section. As suggested by the reviewer, the authors have improved the clarity and structure of the paper overall by re-writing the introduction.

First, the research gap is now formulated more clearly in the introduction:

“But when it comes to bio-inspiration, the question of scalability of natural structures becomes common. For instance, a closed shell, such as an avian egg, can rest on a plane without being damaged at a small scale. However, when scaled up the shell’s self-weight and thus the impact of gravity increases. Under the same support conditions as the small-scale structure, the large upscaled shell could be subject to localized deformation such as buckling [7]. While this action of gravity is easily understood in this example of an egg-like shell, the question remains open to determine at which scale the action of gravity becomes too great for compliant shells to operate reliably. Shell structures span over 10 orders of magnitude across both biology and engineering. Shell mechanics are used to describe the large shape transitions of viruses [8], and of red blood cells [9], and are the mechanical system for some of the fastest repeatable plant movements [1]. Their flexibility allows the movement in engineered compliant structures such as hingeless joints [10] or active piezoelectric actuators [11]. All these flexible structures are elastically deformed, which makes them susceptible to undergo large stresses. However, similarly to their stiff counterparts, the geometry of compliant shells influences the magnitude of those stresses [12]. With advances in the modeling of large deformations [13, 14], the use of shells as mechanisms is now made possible. While biological compliant shells appear at the smallest of the 10 length scales cited above, the use of those structures in biology has started to inform the design of engineered mechanisms at larger scales. Examples of flexible shells observed in nature have inspired engineered scale adaptive structures [15-18] at the meter scale, but the question of whether such structures could be scaled up even further still remains open and drives this study. This succinct literature review shows that there is a clear gap of knowledge as to what the limiting scale of compliant shells is and what the defining parameter is that determines this scale.

Second, the research goal and the four research tasks of the article have been defined more clearly:

“The main hypothesis guiding this study is that the lack of large-scale compliant shells is due to the limiting effect of gravity-induced body forces on the shell’s movements. Therefor the goal of this paper is to gain insight in the influence of gravity-induced forces on the ability of shells to perform as mechanisms through an order-of-magnitude approach. To achieve this goal, we are guided by the following four research tasks. First, we identify and catalogue the dimensions and mechanical characteristics of shell instances across 10 orders of magnitude of span. Second, we apply the non-dimensional Föppl-von-Kármán number [19] to each of those shells to characterize the most likely deformation mode (i.e. bending or stretching). Third, in order to characterize the influence of gravity forces on a shell, we introduce a new non-dimensional number called the gravity impact number (Gi), which is the ratio of the elastogravity length scale [20, 21] to the characteristic dimension of the shell. The elastogravity length scale determines the limit at which bending deformations due to gravity appear in the shell. Finally, using this newly introduced parameter, we measure the scale at which compliant shells become highly susceptible to gravity induced deformations.”

Third, the research contributions are now clearly stated:

“The main contributions of the paper are 1) the cataloguing of stiff and compliant shells across scales of 10 orders of magnitude, 2) the use the Föppl-von-Kármán number to characterize the mechanical behavior of those shells, 3) the introduction of the gravity impact number to describe the scale at which the pull of gravity becomes a dominant factor in the analysis of shells and 4) the determination  that this scale is ~0.1 m.”

How are the flexibility characteristics implemented in the physical model? Explain more

The study is a dimensional analysis of shells found in literature. 64 instances of both stiff and highly flexible shells found in literature, were identified and selected to compare their mechanical behavior. . . The shells’ss material properties and size are examined in the context of their of elastic deformations, which could either be small (for stiff shells) or large (for flexible shells) . The shell flexural stiffness D is the main shell characteristic that is present in both the Föppl-von-Kármán number and the gravity impact number. For all 64 shell instances used in our study, this flexural stiffness D is available in the appendix.

Have been conceptualized some experiments to validate the theories developed?

In the work of the author “Charpentier, V., Adriaenssens, S., & Baverel, O. (2015). Large displacements and the stiffness of a flexible shell. International Journal of Space Structures, 30(3-4), 287-296" the upscaling of a carnivorous plant was described in detail. The carbon fiber structure mentioned in the paper is 0.8 m long. At the time of the study, the scale was chosen based on manufacturing capabilities, the goal was to build the largest shell possible. The physical shell experienced significant bending due gravity. The initial shape of the structure varies depending on its orientation with respect to gravity (acting along the z-axis). The structure has two types of orientations based on the position of its active rib (line of action of the actuation, see paper for details). If the rib is oriented along the vertical axis (z axis), the trajectories of any point on the shell occur at constant z. Since gravity is carried along the z axis as well, the low stiffness mode of deformation activated during the movement is not impacted by gravity. If the rib belongs to the horizontal plane (x-y plane), the points of the shell have varying z values during actuation: the deformation is impacted by gravity. In this case, when no actuation is applied, gravity produces a force on the shell that flattens the initial shape if the shell closes upward (corresponding to the position in the paper’s experiment) and curves/closes the initial shape if the shell closes downward (opposite to the direction of closure in the paper’s experiment). While the scale of this structure allowed the authors to display large elastic deformation, it made the shell’s deformation highly influenced by gravity loading. This experiment showed that a scale larger than the 0.1 m proposed in this paper is not independent from gravity.

Round 2

Reviewer 1 Report

The authors have answered all the questions of reviewers in an appropriate way. the paper can be published in the current form.